# Biological Activities and Ecological Significance of Fire Ant Venom Alkaloids

**DOI:** 10.3390/toxins15070439

**Published:** 2023-07-03

**Authors:** Guangxin Xu, Li Chen

**Affiliations:** 1School of Life Sciences, Institute of Life Science and Green Development, Hebei University, Baoding 071002, China; jainxuguang@163.com; 2Hebei Basic Science Center for Biotic Interaction, Hebei University, Baoding 071002, China

**Keywords:** *Solenopsis invicta*, venom gland, piperidine, piperideine, venom toxicity, antimicrobial activity, ecological effect

## Abstract

Venoms produced by arthropods act as chemical weapons to paralyze prey or deter competitors. The utilization of venom is an essential feature in the biology and ecology of venomous arthropods. *Solenopsis* fire ants (Hymenoptera: Formicidae) are medically important venomous ants. They have acquired different patterns of venom use to maximize their competitive advantages rendered by the venom when facing different challenges. The major components of fire ant venom are piperidine alkaloids, which have strong insecticidal and antibiotic activities. The alkaloids protect fire ants from pathogens over the course of their lives and can be used to defend them from predators and competitors. They are also utilized by some of the fire ants’ natural enemies, such as phorid flies to locate host ants. Collectively, these ants’ diverse alkaloid compositions and functions have ecological significance for their survival, successful invasion, and rapid range expansion. The venom alkaloids with powerful biological activities may have played an important role in shaping the assembly of communities in both native and introduced ranges.

## 1. Introduction

Venom is a toxic secretion produced in animals by specialized glands that are often connected to teeth, stingers, or other sharp organs and delivered to target organisms to disrupt their normal physiological or biochemical processes [1,2]. The animals that utilize highly toxic venom for predation, defense, and competitor deterrence are believed to possess evolutionary advantages [2,3,4]. Among the venomous taxa (including cnidarians, echinoderms, mollusks, vertebrates and arthropods encompassing ants, bees, wasps, spiders, and scorpions), ants are the most abundant social insects and one of the most successful animals in number and geographical range in the world [5], having complex social organization and being found in a variety of ecosystems around the planet. In the ant community, venoms are chemically diverse and used by venomous ants as a chemical weapon to protect their own nests or to seize food resources from competitors [6].

Fire ants in the genus *Solenopsis* (Hymenoptera: Formicidae) are venomous and aggressive, and their common name “fire ants” derives from the burning sensation and pain felt by humans after being stung. Through their stingers, these ants secrete a venom in which major components are piperidine alkaloids (over 95%) and minor components are water-soluble proteins (less than 1%) [7,8,9,10,11]. The composition of alkaloids in the venom of different *Solenopsis* species are qualitatively and quantitatively different [12,13], which is an important feature in *Solenopsis* taxonomy, especially within the *Solenopsis saevissima* species complex that exhibits the greatest diversity of piperidine components [14,15,16,17]. The more basal *Solenopsis* species (e.g., *Solenopsis geminata* and *Solenopsis xyloni*) produce simpler piperidines compared to more derived lineages (e.g., *Solenopsis richteri* and *Solenopsis invicta*), which produce more complex analogs [18].

As an invasive ant, *S. invicta* has invaded multiple areas in the world [19]. The invaded populations are ecologically dominant, often reduce native ant diversity, and have negative impacts on other organisms in both agricultural and natural ecosystems [20,21,22]. Possessing chemical weaponry is one of the important factors that contribute to the success of *S. invicta*. The alkaloids in the venom of fire ants are used to hunt, to defeat their opponents, or as a defensive measure against pathogens [23,24,25,26,27]. The effective utilization of venom by several fire ants, such as *S. geminata*, *S. richteri*, and *S. invicta*, may have facilitated their successful invasions and rapid spread to multiple areas, where they establish new populations, or even replace the dominant native ant species [28,29,30,31]. In the text below, “fire ant” refers to *S. invicta*, unless otherwise specified. In this review, we focus on (a) patterns of venom use by fire ants, (b) chemical structure, (c) biological activities, and (d) the ecological significance of fire ant venom alkaloids.

## 2. Patterns of Venom Use in Fire Ants

Fire ant venom apparatus (Figure 1) is comprised of the stinger, venom sac, venom gland, and Dufour’s gland [32]. The venom gland connects with the venom sac, and its main function is to synthesize the venom components [32,33,34] that are stored in the venom sac. Dufour’s gland is an accessory gland consisting of a single row of large cuboidal cells, which produce trail pheromones [32,35]. The muscles around the base of the stinger contract to control the back and forth movement of the stinger to penetrate the skin of attacked animals [32]. Meanwhile, at the base of the sting bulb, there is a valve connecting to the muscles to operate the open-up or shut-off mechanisms of the venom sac. The backward and forward motion of the valve helps to force the venom to the stinger tip and into the wound of the attacked animal [32]. However, there are no muscle fibers associated with the venom sac. Thus, the propelling force of the venom fluid’s injection has been proposed to be provided by a strong contraction of the gaster [36].

Three patterns of venom use have been observed. First, when disturbed by human beings or other large animals, a fire ant worker uses its mandibles to clamp onto a soft body part of a victim and then injects venom into the body through its stinger (Figure 2A). After one or multiple stings at one site, it loosens its mandibles and moves to other sites for repeated attacks. When subduing prey such as caterpillars and other small animals, fire ant workers inject venom in a similar manner. The average volume of a sting is about 0.66 nL, equivalent to 3.1% of the total venom (per individual) [37]. Lai et al. (2010) reported that the average venom dose per sting of workers with head widths ranging from 0.75 to 0.97 mm is 1.01 nL for monogyne *S. invicta*, 0.79 nL for polygyne *S. invicta*, and 0.74 nL for *S. geminata* [24]. However, the size and age of a worker can impact the amount of venom delivered per sting. The venom dose per sting increases with age until one month old but then declines after a mid-age peak [38]. This strategy could be due to the transition of a worker ant from the responsibility of brood care and nest maintenance to nest defense and foraging, as well as a possible reflection of a positive relationship between venom dose and venom reserve in the worker ant. Older workers (foragers) deliver lower venom doses than mid-age workers (nurses and resource defenders) to slow the depletion of their limited venom supplies because they are unable to synthesize appreciable amounts of venom to replace spent venom [37].

Second, when competing with other species of ants, a fire ant worker elevates its gaster, extrudes the stinger, vibrates the gaster, and secretes a drop of venom at the stinger tip, showing a typical defensive gaster flagging behavior (Figure 2B) [23]. The venom droplets may be dispersed through the rapid vibration of the gaster or dabbed onto the body surface of other ants through direct contact [23,39]. An average venom dose dispersed during gaster flagging (0.50 μg) is slightly lower than that from a sting (0.56 μg) [23,37] but slightly higher than the LD_50_ value (0.489 μg) of fire ant venom against an Argentine ant, *Linepithema humile* [25], suggesting that fire ant venom dispersed through gaster flagging and dabbing behavior is a potent weapon against competing ants. Although it is fatal for most competing ants [25], the fire ant venom can be efficiently detoxified by the venom of the tawny crazy ant, *Nylanderia fulva* [39]. Consequently, scattered colonies of the tawny crazy ants in some regions of the Texas Gulf Coast have displaced most *S. invicta* colonies where they have met [40]. However, gaster-flagging behavior in the most studied fire ant species other than *S. invicta*, such as *S. geminata* and *S. saevissima*, has not been described. Ants of the genus *Monomorium*, which produce venom consisting mostly of pyrrolidines [14,41], also display gaster flagging and dabbing behavior [42]. *Megalomyrmex* thief ants frequently dispense venom as an aerosol through gaster flagging or as a drop by dabbing their gaster on heterospecifics [43]. These venom-dispensing behaviors seem to be widely shared by all stinging ants [23,42].

Third, brood-tending workers dispense a small quantity of venom to the brood surface (~1 ng) or the surrounding brood chamber by vibrating or repeatedly shaking their gasters. Presumably, the thin film of venom functions here as an antibiotic to reduce the likelihood of microbial infection [23].

## 3. Alkaloids in Fire Ant Venom

Alkaloids in fire ant venom are derivatives of 2-methyl-6-alkylpiperidines and 2-methyl-6-alkenylpiperidines [44,45]. The length of their side carbon chain at the 6-position of the piperidine ring ranges from 7 to 17 carbons [16]. The two carbon atoms at the 2- and 6-positions of the piperidine ring are chiral. Thus, the absolute configuration of an alkaloid component can be (2*R*,6*R*), (2*R*,6*S*), (2*S*,6*R*), and (2*S*,6*S*) [46,47,48]. Most studied *Solenopsis* species produce a single enantiomer of both the *cis* (2*R*,6*S*) and *trans* (2*R*,6*R*) forms of each piperidine alkaloid. The other two enantiomers, *cis* (2*S*,6*R*) and *trans* (2*S*,6*S*), are only found in the Brazilian fire ant *S. saevissima* [46,48]. Absolute configuration of these piperidines is a useful tool to differentiate sympatric *Solenopsis* species from *S. saevissima* [48]. Various structures of the alkaloids in the fire ant venom are listed in Table 1.

Since the first identification of the piperideine alkaloid *Δ*^1,2^-C11 from *Solenopsis xyloni* [49], a series of *Δ*^1,2^- and *Δ*^1,6^-piperideine alkaloids (Table 1) have been reported from *Solenopsis* fire ant venom [13,15,16,50,51,52]. The two *Δ*^1,2^- and *Δ*^1,6^-C11 piperideines can be thermochemically reduced to *cis*-C11 and *trans*-C11 at a ratio of 4:1 [53,54]. This ratio of the equilibrium mixture of *cis*- and *trans*-C11 formed during their chemical synthesis matches that of these two major components (*cis*- and *trans*-C11) in the venom of *S. aurea* [55]. This match-up supports the hypothesis that *Δ*^1,2^- and *Δ*^1,6^-piperideines function as precursors for the biosynthesis of piperidine alkaloids in the fire ant venom [13,50,56,57]. The two *Solenopsis* fire ants in the *saevissima* species complex, *S. richteri* and *S. invicta*, and their hybrid, *S. richteri* × *S. invicta*, produce predominant *trans* alkaloids. Enantioselective enzymes present in these species have been proposed to reduce *Δ*^1,2^-piperideines exclusively into (2*R*,6*R*)-dialkylpiperidines, and *Δ*^1,6^-piperideines mainly into (2*R*,6*R*)-dialkylpiperidines and partially into (2*R*,6*S*)-dialkylpiperidines [50]. Because *Δ*^1,2^-piperideines and corresponding *Δ*^1,6^-piperideines are quantitatively indistinguishable, the two reduction routes may be equally important in the biosynthesis of *trans* alkaloids in the imported fire ants [50]. However, the fact that the ratio of *cis*-C11 to *trans*-C11 is about 1:2 in *S. geminata*, [58] suggests that the *Δ*^1,2^-C11 pathway (enantioselectively reduced to (2*R*,6*R*)-*trans*-C11) could be more important than the *Δ*^1,6^-C11 pathway (enantioselectively reduced to (2*R*,6*S*)-*cis*-C11) in more ancient *Solenopsis* species. Since all four stereoisomers are present in *S. saevissima* [46,48], the enzymes used to catalyze the reduction of *Δ*^1,2^-piperideines and *Δ*^1,6^-piperideines may lack enantioselectivity. It seems likely that *Solenopsis* fire ants utilize similar enzymes with very different enantioselectivity to synthesize both *cis* and *trans* alkaloids. The New World *Solenopsis* fire ants may have evolved specific enzymes to synthesize piperidines with favorable stereochemical composition and longer and unsaturated side carbon chains associated with biological advantages [17,50,57].

In addition to piperidine and piperideine alkaloids, ten pyridine alkaloids (Table 1) have recently been found to be present in trace amounts in *S. invicta* venom [59]. Structures of the three pyridines with a saturated side carbon chain, including 2-methyl-6-undecylpyridine (2M6UP11), 2-methyl-6-tridecylpyridine (2M6TP13), and 2-methyl-6-pentadecylpyridine (2M6PP15), have been confirmed through comparisons with synthetic compounds. The three pyridines with an unsaturated side carbon chain, including two isomers of 2-methyl-6-tridecenylpyridine (2M6TP13:1) and 2-methyl-6-pentadecenylpyridine (2M6PP15:1), were tentatively identified with mass spectra [59]. Saturated/unsaturated pyridine alkaloids are also present in *Solenopsis geminata* venom, including 6-undecylpyridine, 2-methyl-6undecyl-pyridine, and 2-methyl-6-(1)-undecenylpyridine [60]. Unfortunately, the biological function and ecological significance of these trace components, except for 2M6UP11 (see below), remain unknown.

## 4. Biological Activities of Piperidine Alkaloids from Fire Ant Venom

The piperidine alkaloids from fire ant venom possess diverse biological activities. Although severe clinical symptoms, such as pain, itching, and anaphylactic reactions, caused by fire ant venom alkaloids have been extensively investigated and attracted considerable attention [61,62], we mainly focus here on their insecticidal and antimicrobial activities, as these are the most important for fire ant survival.

### 4.1. Insecticidal Activities

The chemical nature and insecticidal activities of fire ant venom were first discovered in late 1950s [63]. The residue and topical application of fire ant venom demonstrated that the venom was highly toxic to fruit fly (*Drosophila melanogaster*), housefly (*Musca domestica*), termite (*Kaleotermes* sp.), boll weevil (*Anthonomus grandis*), rice weevil (*Sitophilus oryza*), and mites (*Tetranychus telarius* and *T. cinnabarinus*). The treated insects exhibited rapid paralytic symptoms that were highly suggestive of a nerve antagonistic effect [63].

Fire ants often use their defensive venom against other sympatric ant species during resource competition. For example, under laboratory conditions, they use gaster flagging and venom dabbing behaviors to apply enough venom to be lethal to Argentine ants during Argentine ant−fire ant interactions. The susceptibilities of a number of ant species from California to fire ant venom are highly variable. When adjusted for the weight of each ant species, the order of decreasing susceptibility to the fire ant venom is as follows: *L. humile* > *Dorymyrmex bicolor* = *Pogonomyrmex californicus* > *Liometopum occidentale* > *S. xyloni* = *Formica perpilosa* > *S. invicta* [25]. Among these ant species, *L. humile* is most susceptible to the venom, whereas *S. invicta* is the most resistant to its own venom. The difference in susceptibility between these two species is 330 times. The high susceptibility of *L. humile* may help explain the superior interference competition posed by *S. invicta*.

*Solenopsis* fire ant venoms have relatively high insecticidal activities against caterpillars. When stung by fire ants once in the dorsal thoracic region or manually dabbed by venom droplets from the tip of the fire ant sting onto the dorsal thoracic region, the larvae of *Spodoptera litura* and *Plutella xylostella* show immediate contractions, flaccid paralysis, black coloration, and death. The fire ant venom-induced darkening may result from melanin accumulation in the integument and hemolymph [64]. Young larvae are apparently more susceptible to fire ant stings and have higher mortality than old larvae. The toxic effects of fire ant stings (LT_50_ values) are ranked as follows: *S. geminata* > monogyne *S. invicta* > polygyne *S. invicta* [24,65]. Since the average venom dose per sting of monogyne *S. invicta* is higher than that of polygyne *S. invicta* and *S. geminata*, the volume of venom from monogyne *S. invicta* applied to the thoracic region of the *P. xylostella* larva should be more than that of *S. geminata* and polygyne *S. invicta*. The significantly lower LT_50_ value of *S. geminata* venom than that of the venoms of both *S. invicta* social forms indicates that the toxicity of *S. geminata* venom is significantly higher than that of both social forms of *S. invicta*. Furthermore, the 35.8% lower LT_50_ of venom derived from monogyne *S. invicta* than that of polygyne *S. invicta* correlates to the 27.8% higher volume of venom derived from monogyne *S. invicta* than that from polygyne *S. invicta*, suggesting that the toxicity of venom from monogyne *S. invicta* against *P. xylostella* is likely higher than that of polygyne *S. invicta* [65]. Because there are wide variations in the amount of venom from different sized workers in the same species delivered to the caterpillars, each caterpillar may receive different amounts of venom, which impacts the accuracy of the venom toxicity data in the above two studies [24,65]. Different fire ant species apparently deliver different amounts of venom to the caterpillars [24]. The reported order of the toxicity of different venoms may lack reliability because of the interspecific variation in the amount of venom applied to the caterpillars [24,65]. Future evaluations of different fire ant venoms should control for exact same dose measurements across fire ant species.

The venoms of *Solenopsis* fire ants constitute mainly defensive 2-methyl-6-alkylpiperidine and 2-methyl-6-alkenylpiperidine alkaloids responsible for their insecticidal activity, and these are considerably different from the constituents of proteinaceous venoms [8,11,25]. There are significant differences in alkaloidal constituents between the three *Solenopsis* species native to the United States (*Solenopsis aurea*, *S. xyloni*, and *S. geminata*) and the imported fire ant *S. invicta*. The major components in the venoms of the three native species are *cis*-C11 and *trans*-C11, and the ratios of *cis*-C11 to *trans*-C11 are about 4:1, 4:1, and 1:2, respectively [17,55,58,66,67,68]. By contrast, these two major alkaloids from native fire ants are minor components in *S. invicta* venom [69]. The *S. invicta* venom contains distinct *trans*-C13, *trans*-C13:1, *trans*-C15, and *trans*-C15:1 that are minor components or absent in *S. geminata* venom [13]. The higher toxicity of *S. geminata* venom against caterpillars compared to that of *S. invicta* venom suggests that saturated piperidines with a short side carbon chain are relatively more toxic than those with a long side carbon chain. However, this suggestion contradicts the finding that the toxicities of both *cis*- and *trans*-C11 against a *Reticulitermes* termite are approximately similar to those of *cis*- and *trans*-C13, *trans*-C15, and nicotine, but somewhat higher than that of *cis*-C15 (Table 2) [70]. The toxicity of synthetic *trans*-C11 against Argentine ants is the same as that of *S. invicta* venom. However, it is surprising that both *S. invicta* and *S. xyloni* are six times more susceptible to synthetic *trans*-C11 than to *S. invicta* venom, meaning that synthetic *trans*-C11 is more toxic than *S. invicta* venom to the two fire ants [25]. Fox et al. found that *S. invicta* venom is more lethal than *S. geminata* venom against sympatric competitor ant species [69]. The reported higher toxicity of *S. geminata* venom versus *S. invicta* venom against caterpillars [24,65] could be due to the higher observed speed of knockdown incapacitation by *cis*- and *trans*-C11 than *trans*-C13 and *trans*-C15 (Table 2) [69]. Between the two social forms of *S. invicta*, there exists a significant difference in the ratios of *trans*-C13:*trans*-C13:1 and *trans*-C15:*trans*-C15:1 [67]. The proportions of unsaturated alkaloids in the venom of polygyne workers are significantly higher than the corresponding proportions in monogyne workers. This difference may account for the lower toxicity of venom from polygyne *S. invicta* than that from monogyne *S. invicta* against *S. litura* and *P. xylostella* larvae. However, the lower toxicity of polygyne *S. invicta* venom containing higher proportions of unsaturated alkaloids does not agree with the finding that the alkaloids with a double bond in the side carbon chain are as toxic or more toxic than their saturated counterparts [70].

Both piperidine and piperideine alkaloids from the fire ant venom are toxic to green peach aphids, *Myzus persicase*. These alkaloids act rapidly and cause death in green peach aphids in as few as four hours [71]. Furthermore, the insecticidal activities of the two classes of alkaloids do not significantly differ; however, the elongation of the side carbon chain of the piperideines decreases insecticidal activity (Table 2) [72]. These two classes of alkaloids offer potential for the development of new insecticides as alternatives to environmentally harmful synthetic insecticides.

Interestingly, fire ant venoms show repellant activity to competing ant species that do not synthesize alkaloids. For instance, Argentine ants and odorous house ants, *Tapinoma sessile*, are repelled by food treated with fire ant alkaloids. However, ant species that produce alkaloidal venoms (*Solenopsis* and *Monomorium* species) are not deterred by food containing the alkaloids [73]. Both 2-methyl-6-alkylpiperidines and 2-methyl-6-alkenylpiperidines are effective repellents, especially those with a shorter side carbon chain. In general, these piperidines, with a *cis* configuration, are a better deterrent than the *trans*-isomers. Because the alkaloids in *S. invicta* venom are mainly *trans*-isomers, *S. invicta* venom is apparently a less effective deterrent than venoms of other fire ant species [70].

### 4.2. Bactericidal Activities

The antibacterial activities of fire ant venom have been reported since 1958 [63]. Paper discs impregnated with a 1:50 dilution of natural venom proved to effectively inhibit the growth of *Micrococcus pyogenes*, *Streptococcus pyogenes*, *Escherichia coli*, and *Lactobacillus casei* on agar plates [63]. Synthetic fire ant venom alkaloids were also tested for antibacterial properties against a variety of bacteria using disc-diffusion procedures [74]. Gram-positive bacteria were found to be more sensitive to *trans*-C11, *trans*-C13, and *trans*-C15 than Gram-negative ones. Furthermore, the order of inhibition on the gram-positive bacteria was always the same (*trans*-C11 > *trans*-C13 > *trans*-C15), i.e., the activity of these piperidines tended to decrease with the increase in length of the side carbon chain at the 6-position of the piperidine ring. Because these synthetic piperidines acidified by HCl were directly dissolved in water to prepare the 1:1000 aqueous solution, the solubility of piperidine HCl might significantly impact their inhibitory abilities. The higher antibacterial activity of *trans*-C11 compared with *trans*-C13 and *trans*-C15 in the disc-diffusion tests might result from its higher solubility (maximum concentration of the aqueous solution: 250–500 μg/mL for *trans*-C11 HC1, 125–250 μg/mL for *trans*-C13 HCl, and 10–20 μg/mL for *trans*-C15 HCl). Therefore, the order of *trans*-C11 > *trans*-C13 > *trans*-C15 may not be correct. Under the same concentration, for example, *trans*-C11 was found to be less effective than *trans*-C13 and *trans*-C15 against *Staphylococcus aureus*, and less effective than *trans*-C13 against *E. coli* (Table 2) [74]. However, *trans*-C13:1 was ineffective against all bacteria tested [74].

The antibacterial activities of nine synthetic fire ant venom alkaloids, *cis*-C11, (2*S*,6*R*)-*cis*-C11, (2*R*,6*S*)-*cis*-C11, *cis*-C13, (2*S*,6*R*)-*cis*-C13, (2*R*,6*S*)-*cis*-C13, *trans*-C11, (2*R*,6*R*)-*trans*-C11, and (2*S*,6*S*)-*trans*-C13, have been further evaluated using the broth dilution method against six species of bacteria (*Streptococcus pneumoniae*, *Staphylococcus aureus*, *Enterococcus faecalis*, *Escherichia coli*, *Stenotrophomonas maltophilia*, and *Pseudomonas aeruginosa*). All the alkaloids inhibit *S. pneumoniae* with minimum inhibitory concentration (MIC) values ranging from 1 to 4 mg/L; however, none of them inhibits *E. coli* or *P. aeruginosa*. *Staphylococcus aureus, E. faecalis,* and *S. maltophilia* are selectively sensitive to *cis*-C13, (2*S*,6*R*)-*cis*-C13, (2*R*,6*S*)-*cis*-C13, and (2*S*,6*S*)-*trans*-C13, having very similar MIC values. These four alkaloids show bactericidal activity against *S. pneumoniae* and *S. aureus* [26].

*Clavibacter michiganensis* subsp. *michiganensis* (CMM) is a Gram-positive bacterium that causes tomato bacterial canker, a highly destructive disease of tomato. Both piperidine and piperideine alkaloids extracted from fire ants show strong inhibitory activity against CMM, supporting the findings reported by Jouvenaz et al. [74]. The growth of CMM was negatively correlated with the concentration of piperidine alkaloids in nutrient broth. It seems likely that piperidine alkaloids kill CMM directly or inhibit the cell division of CMM. Piperideine alkaloids at two application concentrations (18.8 and 188.5 μg/mL) inhibit the symptom development of the bacterial canker on tomato seedlings in greenhouse conditions [27]. Moreover, the inhibitory activity of piperidine alkaloids remains quite stable at room temperature (22 °C) for a storage period of 12 weeks and at 54 °C for up to 4 weeks. The stability of piperidine alkaloids and the effectiveness of piperideine alkaloids in managing plant-pathogenic bacteria warrant further exploration for potential commercialized product development.

Pyocyanin is a quorum-sensing-controlled virulence factor of *P. aeruginosa*. Exposure to synthetic *trans*-C11 dissolved in growth medium inhibits pyocyanin production, indicating a role in suppressing the quorum-sensing signaling of *P. aeruginosa* [75]. This gram-negative bacterium possesses an LuxR/I-type “*rhl*” quorum-sensing signaling system that uses an acyl-homoserine lactone autoinducer, C4-homoserine lactone. Quorum sensing in *P. aeruginosa* can be restored using the exogenously added synthetic C4-homoserine lactone, suggesting that *trans*-C11 targets the *rhl* signaling system [75]. Elastase B is a metalloprotease that promotes pathogenesis. Its production is partly controlled by C4-homoserine lactone-induced quorum-sensing signaling. Moreover, *trans*-C11 also suppresses elastase B production; however, the effect is not as dramatic as that on pyocyanin production. These findings strongly suggest that *trans*-C11 acts mainly on the *rhl* quorum-sensing system as a competitor of C4-homoserine lactone [75].

Biofilm formation by microorganisms on exposed surfaces is a major health and industrial concern [76]. *Pseudomonas fluorescens* is one of the most intensively studied gram-negative bacteria that can form biofilms on different surfaces, such as polystyrene, stainless steel, and polyamides [77]. An extract of fire ant venom alkaloids can reduce adhesion and biofilm formation through the growth of *P. fluorescens* on polystyrene and stainless-steel surfaces, exhibiting potential application as a surface conditioning agent against biofilm formation [78]. Synthetic *trans*-C11 is able to reduce the biofilm formation in *P. aeruginosa* in a dose-dependent manner [75]. This biofilm formation is also regulated by quorum-sensing signaling [79]. In addition to an inhibition of the production of pyocyanin and elastase B, suppressed biofilm formation accounts for the *rhl* quorum-sensing signaling in *P. aeruginos* that is disrupted by *trans*-C11. The fire ant venom alkaloids are therefore lead compounds for the development of therapeutic *rhl* signaling inhibitors.

2-Methyl-6-alkyl-*Δ*^1,6^-piperideine (*Δ*^1,6^-C15) shows antibacterial activity against vancomycin-resistant *Enterococcus faecium*. Its 50% growth inhibition (IC_50_) and MIC values are 19.4 and 20.0 μg/mL, respectively. Two analogues of 2-methyl-6-alkyl-*Δ*^1,6^-piperideines, *Δ*^1,6^-C14 and *Δ*^1,6^-C16, are inactive against this bacterium [80].

### 4.3. Fungicidal Activity

Fire ant venom was first demonstrated to possess fungicidal activities as early as the 1950s; however, the report only mentioned that a variety of molds were inhibited by a 1:50 dilution of natural venom [63]. Since then, fire ant venom alkaloid extract, purified natural and synthetic venom alkaloids from *Solenopsis* fire ants have been shown to possess pronounced fungistatic and/or fungicidal activities against various fungi.

The venom alkaloid extract of hybrid fire ants (*S. richteri* × *S. invicta*) prevents the growth of plant fungal pathogens, such as *Botrytis cinerea*, *Fusarium oxysporum*, *Phytophthora nicotianae*, *P. cryptogea*, *Phytopythium citrinum*, *Rhizoctonia solani*, and *Sclerotonia rolfsii* [81]. Fire ant venom alkaloids incorporated in both solid and liquid cultures inhibit the conidial germination of *Beauveria bassiana* isolates, AF-4 and 447, *Metarhizium anisopliae,* and *Paecilomyces fumosoroseus*. They act fungistatically on *B. bassiana* and *M. anisopliae*. The order of conidia sensitivity to the venom alkaloids is *P. fumosoroseus* > *M. anisopliae* > *B. bassiana* 447 isolate > *B. bassiana* AF-4 isolate [82]. The alkaloids at relatively high concentrations in liquid media also elicit a hyphal body formation that is usually dependent upon temperature, nutrition, or a combination of the two. Because cellular lysis of either hyphal bodies or mycelia was not found following the germination of conidia, the effect of the venom alkaloids on the fungus was thought to only affect conidial germination [82]. Further research should aim to explore the exact mechanism involved in cell wall synthesis interference by the venom alkaloids.

The pure cultures of 13 fungal species grown on potato dextrose agar were evaluated for toxicity with synthetic venom alkaloids (*cis*-C9 + *trans*-C9 (85:15), *cis*-C11, *cis*-C13, *cis*-C13:1, *cis*-C15, *cis*-C15:1, *trans*-C11, *trans*-C13, *trans*-C13:1, *trans*-C15, and *trans*-C15:1) in the 1980s, including three animal pathogens, *Trichophyton rubrum*, *T. mentagrophytes*, and *Microsporum canis*; two plant pathogens, *Pythium irregulare* and *Rhizoctonia solani*; five species isolated from *Forelius pruinosus* larvae, *Fusarium oxysporum*, *Cunninghamella echinulata*, *Gliocladium deliquescens*, *Penicillium* sp., and *Paecilomyces marguandii*; and three species isolated from *S. invicta* larvae, *Aspergillus zonatus*, *Zygorhynchus vuilleminii*, and *Mucor* sp. [70]. All these fungal species show great variability in their growth responses to the synthetic alkaloids. In general, all the synthetic alkaloids exhibit effective inhibition against the 13 species except for the three ant-derived species, *C. echinulata*, *Penicillium* sp., and *A. zonatus*. The *cis*-isomers of C11, C13, C13:1, C15, and C15:1 are equivalent in activity to, or slightly less effective than, the corresponding *trans*-isomers as fungal growth inhibitors against the 13 species (Table 2). Among the 11 venom alkaloids, the mixture of *cis*-C9 and *trans*-C9 (85:15), *cis*-C11, and *trans*-C11 are the most active inhibitors of fungal growth. For each series of alkaloids under the same configuration (*cis*-piperidine with saturated side carbon chain, *trans*-piperidine with saturated side carbon chain, *cis*-piperidine with unsaturated side carbon chain, *trans*-piperidine with unsaturated side carbon chain), the fungicidal activity tends to decrease with an increase in the length of the side carbon chain (Table 2). For those alkaloids with the same side carbon chain length and under the same configuration (the *cis*-isomers with different side carbon chain saturation status versus the *trans*-isomers with different side carbon chain saturation status, i.e., piperidine with 6-alkenyl side carbon chain versus piperidine with 6-alkyl side carbon chain), the addition of a double bond in the side carbon chain appears to increase the fungicidal activity, i.e., piperidines with a 6-alkenyl side chain are generally more inhibitory than their saturated counterparts (Table 2) [70,83]. 2-Methyl-6-undecylpyridine is generally less active than the other alkaloids for 8 of the 13 species. Moreover, it is about as active as its saturated counterparts, piperidines *cis*-C11 and *trans*-C11, against *R. solani*, *C. echinulata*, *A. zonatus*, and *Penicillium* sp. [70].

*Pythium ultimum* is a soilborne pathogen on crop plants, causing significant yield losses. Both piperidine and piperideine alkaloids extracted from *S. invicta* workers inhibit the mycelium growth of *P. ultimum*. Sporangia are more sensitive than mycelium to piperidine alkaloids. The inhibitory activities of piperidine alkaloids against *P. ultimum* are stable during storage at both 22 °C and 54 °C. A drench treatment with 56.5 μg of piperideine alkaloids per container inhibits both mycelium growth and sporangium germination of *P. ultimum* and improves seedling emergence and the growth of cucumber in greenhouse conditions [84]. Therefore, piperideine alkaloids can be employed to control cucumber loss caused by *P. ultimum*.

Piperideines, *Δ*^1,6^-C14 and *Δ*^1,6^-C15 show antifungal activities against the clinically important opportunistic fungal pathogens *Cryptococcus neoformans* and *Candida albicans*, while piperideine *Δ*^1,6^-C16 is only active against *C. neoformans*. *Δ*^1,6^-C15 is more active than *Δ*^1,6^-C14 and *Δ*^1,6^-C16 against these pathogens [80]. In accordance with an earlier report on the antifungal activities of *Δ*^1,6^-piperideines with side carbon chain lengths from C14 to C18 [85], chain length appears to determine the degrees of antifungal activities of *Δ*^1,6^-piperideines.

### 4.4. Anti-Protozoan Activity

Chagas disease, caused by the protozoan parasite *Trypanosoma cruzi* and transmitted by haematophagous triatomine bugs, is an important disease affecting millions of individuals in the New World. The alkaloids extracted from fire ants *Solenopsis saevissima* and *S. invicta* are toxic to *T. cruzi* amastigotes and epimastigotes [86]. They induce morphological alterations in *T. cruzi* epimastigotes, prompting hypertrophied contractile vacuoles and intense cytoplasmic vacuolization. Further induced biochemical alterations include a type of autophagy, programmed cell death, and reduced intracellular proliferation of *T. cruzi* amastigotes. Although different constituents of venom alkaloids exhibit a diverse cytotoxic effect on different bacterial species, venoms of *Solenopsis saevissima* and *S. invicta* that have distinct alkaloidal mixtures of piperidines [87,88] have similar effects against *T. cruzi*. IC_50_ values for the two venoms are similar against the epimastigote and amastigote forms of *T. cruzi*. The growth inhibitory effects of fire ant venom alkaloids on *T. cruzi* epimastigotes are reversible [86].

African trypanosomiasis, also known as “sleeping sickness”, is caused by the flagellate protozoan *Trypanosoma brucei* in sub-Saharan Africa. Piperideines *Δ*^1,6^-C14, *Δ*^1,6^-C15, and *Δ*^1,6^-C16 strongly inhibit *T. brucei*, with IC_50_ values in the range of 2.7–4.0 μg/mL, which are better than alpha-difluoromethylornithine, with an IC_50_ value of 6.1 μg/mL. *Leishmania donovani* is an intracellular parasite of humans and other mammals, causing a malaria-like disease. All three compounds also show strong antiprotozoal activity against *L. donovani* promastigotes within the human leukemia monocyte THP-1 cells, with IC_50_ values in the range of 5.0–6.7 μg/mL. IC_50_ values of *Δ*^1,6^-C14 and *Δ*^1,6^-C15 against *L. donovani* amastigotes are 3.4 and 3.1 μg/mL, respectively. However, they do not show an inhibitory effect on THP-1 cells, suggesting selectivity against parasites and human cells [80].

## 5. Ecological Significance of Venom Alkaloids from Fire Ants

The antibiotic alkaloids in fire ant venom are extensively used by fire ants in various ways, as stated above. Fire ant queens deposit venom alkaloids on eggs to protect them from pathogen infection [89]. Fire ant workers dispense venom on the internal surfaces of their nests, possibly to inhibit the growth of pathogens and to improve the survival rates of the larvae [23]. Fire ant venom alkaloids released in the environment may have ecological consequences in determining community structure. For example, fire ant mound soil has greater fungal abundance but lower species richness and diversity than nearby soil without fire ant occupation [90,91], indicating that some species, likely having symbiotic relationships with fire ants, are tolerant to the venom alkaloids and thrive under alkaloidal conditions [92].

Fire ants usually use venom alkaloids to subdue a wide variety of prey and for resource dominance and defense, having positive (+) or negative (−) impacts on various ecosystems [93,94]. They attack foliage feeding insect pests (+), tend plant-sucking insects (Hemiptera) (−), attack beneficial natural enemies (−), damage crop products (−), and affect wildlife populations (−). Since the alkaloids are lethal to competing ant species [25], fire ants in invaded areas may have pronounced effects on the functional properties of local communities, resulting in functional homogenization across the landscape [95]. Among the factors that contribute to the co-occurrence of sympatric ant species, such as size, moving speed, numerical abundance, and defensive strategies [96], venom’s lethality, determined by its chemistry, appears to be one of the most important strategies utilized [97]. The venom alkaloids confer a competitive advantage to fire ants to exclude native ants from critical food resources [98]. When competing with the Argentine ant under laboratory conditions, fire ant workers apply lethal venom to competing workers via gaster flagging and venom dabbing, showing superior competition by the fire ant. Similarly, greater venom alkaloid toxicity may contribute to the displacement of the native fire ants *S. xyloni* and *S. geminata* over much of their range in the United States by *S. invicta* [25,65,99]. However, the competitive advantages of fire ant venom alkaloids can be compromised by the tawny crazy ant *N. fulva* [39].

Myrmicine ants comprise a large portion of neotropical dendrobatid frog diets [100]. These poison frogs have the potential to sequester alkaloids from their ant diet for antipredator defense [101]. More than 20 2,6-disubstituted piperidine alkaloids, including *cis*-C11, have been found to be sequestered, although poorly, by some dendrobatid species [102]. However, the poison frog *Dendrobates auratus* is incapable of sequestering piperidine alkaloids from *S. invicta* diet into their skin, possibly due to piperidines’ long hydrocarbon side chains [103].

Ant-derived alkaloids can be utilized as effective cues for host location and recognition by natural enemies [88,104]. Phorid flies in the genus *Pseudacteon* (Diptera: Phoridae) are specific parasitoids of ants. The biology, behavior, and population level impact of phorid flies that parasitize ants of the *saevissima* species complex have been extensively reviewed [105,106,107]. Six highly host-specific *Pseudacteon* species imported from South America have been successfully established across most areas infested by the imported fire ants in the United States. These released fly species perform extremely well in locating fire ant workers. In addition to visual and auditory cues, they use defensive chemicals, mainly venom alkaloids, as olfactory cues for fire ant location at a close range [88,104,105]. Female flies lay eggs into live worker ants, and the hatched larvae eventually decapitate the host ants after consuming all their head tissues. Apart from the direct effect of oviposition on their host ants, phorid flies have an indirect population-level impact on the survival of the host ants by suppressing fire ant foraging behavior and weakening their competing vigor relative to other ant species in the community [105].

## 6. Conclusions

Fire ant venom is comprised of alkaloids (major) and proteins (minor). Large amounts of venom alkaloids are available per colony, and each worker’s venom sac has about 10–20 μg of alkaloids [13]. Fire ant venom alkaloids are derivatives of 2-methyl-6-alkylpiperidines, constituting a diverse chemistry of fire ant venom. Fire ant venoms and their alkaloidal components have significant insecticidal, antibacterial, antifungal, and antiprotozoal activities against a wide variety of organisms. The length of the side carbon chain in the piperidine alkaloids and the ratio of saturated to unsaturated alkaloids influence the toxicity of the venom. In general, the synthetic *trans*-isomers tend to be more toxic than the corresponding *cis*-isomers, and 6-alkenyl-piperidines to be more toxic than their saturated counterparts. There seems to be no general tendency for the alkaloids with different side chain lengths; however, it is likely that their lethality depends on their modes of action. The piperidine and piperideine alkaloids have proven to be effective for various diseases’ prevention and pest management. A large-scale synthesis of the alkaloids is needed for an extensive evaluation of additional disease and pest management potential. Further integrative studies that test for the structure–bioactivity interaction of fire ant venom alkaloids are also required. These alkaloids are lead compounds for structural modification and can be used to develop new groups of pesticides. Most microorganisms are sensitive to the alkaloids; however, some pathogens appear to be tolerant [70,82]. Further toxicological research should be conducted on the mechanisms involved in resistance to the alkaloids.

Fire ant venom alkaloids have been found to be effective anti-microbials targeting human diseases. They show ceramide-like effects and anti-angiogenic activity [108], likely having inhibitory effects on the proliferation of tumor cells. Their derivatives are able to restore normal skin homeostasis against human psoriasis, a common autoimmune disease [109]. Therefore, fire ant alkaloids are perhaps useful for the recovery of skin diseases caused by bacteria/fungi infection, which can be exploited in drug discovery and development programs. Further pharmacological studies are needed to gain insight into the mechanism of the efficacy of the fire ant venom alkaloids.

There are different patterns of venom use depending on the challenges that *Solenopsis* fire ants face, such as prey hunting, resource competition, and colony defense. They often disperse venom droplets onto ant body surfaces and into the mound via stinger vibration or gaster flagging. The venom is highly toxic to targeted organisms but less toxic to the fire ant itself. Extensive use of this venom has ecological consequences, such as shaping the microbiota in fire ant mounds and manifesting species richness and abundance in the fire ant communities, which deserve further investigation.

Fire ants are highly dependent on their venoms for survival, which is an important trait under extreme levels of selection. More toxic venoms are selected for *Solenopsis* fire ants, at least in part, to mediate interspecific interactions. Functional diversity of fire ant venom was selected to shape fire ant dominance as well. Hypothesis-driven studies on the ecological role of fire ant venom alkaloids will not only shed light on focal taxa, but also provide a new horizon for the evolution and ecology of *Solenopsis* fire ant venom.

## Figures and Tables

**Figure 1 toxins-15-00439-f001:**
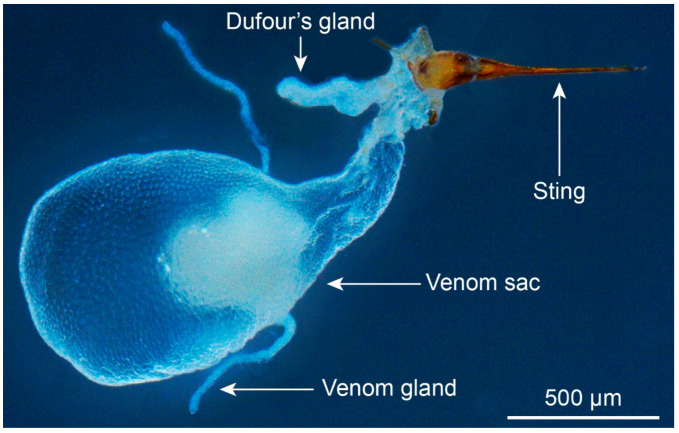
Venom apparatus of *Solenopsis invicta*.

**Figure 2 toxins-15-00439-f002:**
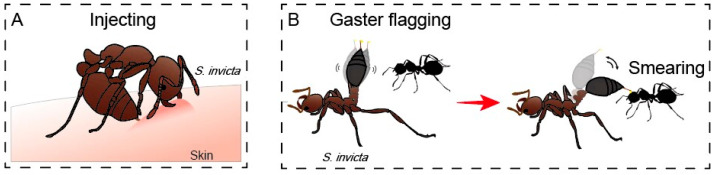
Two commonly observed patterns of venom used by fire ants. (**A**) Injecting pattern; (**B**) Gaster flagging and smearing pattern.

**Table 1 toxins-15-00439-t001:** Alkaloids from *Solenopsis* fire ant venom.

Name	Structure	Name	Structure
*cis* Piperidines		*trans* Piperidines	
*cis*-C7	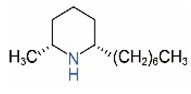		
*cis*-C9:1	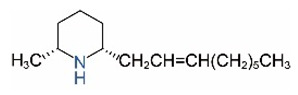		
*cis*-C9	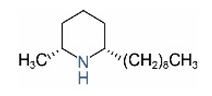		
*cis*-C11:1 ^a^	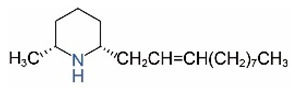	*trans*-C11:1	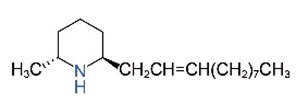
*cis*-C11	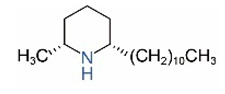	*trans*-C11	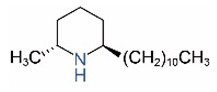
*cis*-C13:1	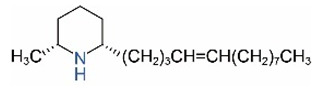	*trans*-C13:1	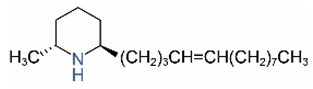
*cis*-C13	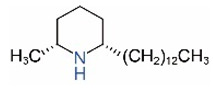	*trans*-C13	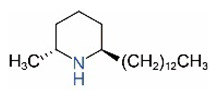
*cis*-C15:1	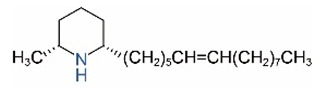	*trans*-C15:1	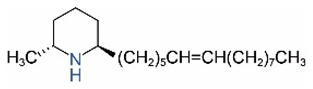
*cis*-C15	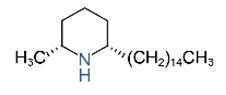	*trans*-C15	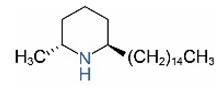
*cis*-C17:1	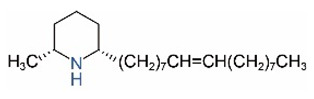	*trans*-C17:1	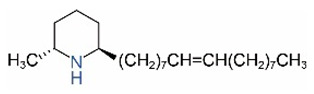
*cis*-C17	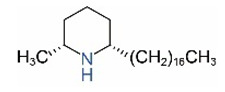	*trans*-C17	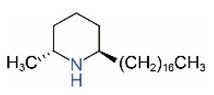
Piperideines			
*Δ*^1,2^-C11	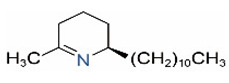	*Δ*^1,6^-C11	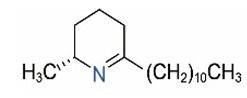
*Δ*^1,2^-C13:1	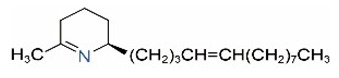	*Δ*^1,6^-C13:1	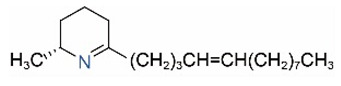
*Δ*^1,2^-C13	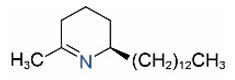	*Δ*^1,6^-C13	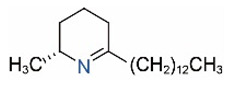
*Δ*^1,2^-C15:1	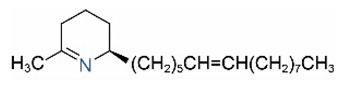	*Δ*^1,6^-C15:1	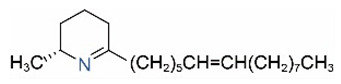
*Δ*^1,2^-C15	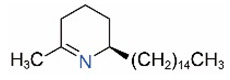	*Δ*^1,6^-C15	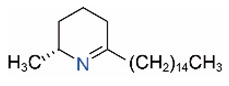
*Δ*^1,2^-C17:1	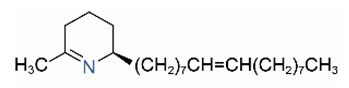	*Δ*^1,6^-C17:1	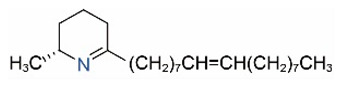
*Δ*^1,2^-C17	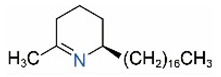	*Δ*^1,6^-C17	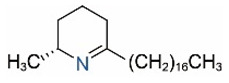
Pyridines			
6UP11	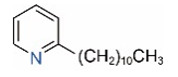		
2M6UP11	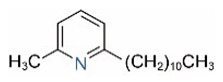	2M6UP11:1	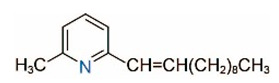
2M6TP13	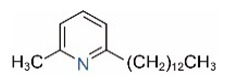	2M6TP13:1	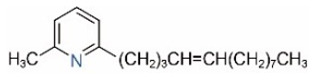
2M6PP15	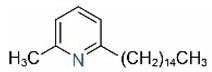	2M6PP15:1	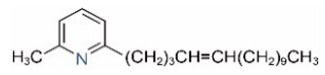

^a^ Carbon numbers are used to represent the trivial names of alkaloids found in *Solenopsis* fire ants, e.g., *cis*-C11:1, with 11 referring to the chain length and 1 referring to the number of double bond. 6UP11: 6-undecyl-pyridine, 2M6UP11: 2-methyl-6-undecylpyridine, 2M6UP11:1: 2-methyl-6-(1)-undecenyl-pyridine, 2M6TP13: 2-methyl-6-tridecylpyridine, 2M6TP13:1: 2-methyl-6-tridecenylpyridine, 2M6PP15: 2-methyl-6-pentadecylpyridine, 2M6PP15:1: 2-methyl-6-pentadecenylpyridine.

**Table 2 toxins-15-00439-t002:** Influence of structural change in fire ant venom alkaloids on their bioactivities.

Structure	Termite	Ant *	Caterpillar	Bacteria	Fungus
Saturated alkaloids, side carbon chain length ↑		↓		↑	↓
Unsaturated alkaloids, side carbon chain length ↑					↓
*cis*-Isomers → *trans*-isomers	↑				↑
Addition of a double bond to the side carbon chain					↑
Piperideine alkaloids, side carbon chain length ↑			↓		

↑: increase; ↓: decrease; →: transition; *: knockdown time.

## Data Availability

The data presented in this study are available in this article.

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
