# Peer review of "Biological Activities and Ecological Significance of Fire Ant Venom Alkaloids"

_toxins, 2023, doi:10.3390/toxins15070439_

Round 1
Reviewer 1 Report
The manuscript “Biological activities and ecological significance of fire ant venom alkaloids” presents a review of the literature on fire at venom chemistry. In general, it provides a worthwhile overview of much of the literature associated with fire ant venom chemistry. It is strongest in its treatment of chemical structure variation in fire ant venom alkaloids and the impact these have on various types of bioactivity. It is weakest in relation to its synthesis and interpretation of the ecological literature. Additionally, it would benefit from an editorial revision. The sentence structure is often overly wordy, and areas of text could be tightened. I am including some specific comments below.
Line #: comment
5: “believe to” is misleading. Fire ant venom definitely acts in these ways.
12: natural enemies
34: “frequently” is unnecessary and misleading (see comment on 5).
35: “more” (see 34)
39: “Tail sting” is used repeatedly throughout and should be eliminated and replaced with stingers. It is unnecessary, inaccurate, and distracting.
45: “to conquier the opponent” overly anthropomorphic
47: “has been considered to facilitate” replace with “facilitates”. These types of sentence contstructions occur throughout. I am not going to highlight them further, but the authors should read and revise.
69: Delete “usually”. Fire ants always bite before stinging.
83-85: This information should be highlighted and moved to earlier in the paragraph. It is the ultimate explanation for the pattern of venom usage described in the paragraph.
93-94: Daubing of venom by fire ants is a potent weapon against any ant not just Argentine ants. Generalize this statement. It makes it appear as if this is a specialized weapon for combat with Argeninte ants.
96-97: No. This is not correct. S. invicta remains the dominant invasive ant species throughout the Gulf Coast. Tawny crazy ants have displaced them from areas they have invaded, but these remain scattered and of small extent on a regional scale.
104: see 47
131-133: As written this sentence implies that venom chemistry varies between the native and introduced ranges of saevissima complex fire ants. Not true. Or at least to my knowledge unstudied. Revise.
142: “primitive” is a loaded term that should be avoided in a science paper.
146-148: There are 191 described species of Solenopsis in the New World. I believe this statement applies only to the 20 or so species of fire ants. Revise.
150: Which fire ant? Clarify.
173 -181: A little more context needed. These are only a small number of ant species from California. Main point is that susceptibility to fire ant venom is highly variable.
180 – 181: The citation is not from a study focused on interefence competition. Lebrun et al 2007 directly studied interference competition using a whole colony removal experiment and found that S. invicta and L. humile were roughly evenly matched with respect to interference competition. Calcaterra 2008 performed a different field study and came to distinct conclusions.
182 – 262: This section on insecticidal activity could be reorganized an greatly tightened.
427: ”infestation” This is a loaded term that is inapproriate in this context.
419 – 461: This section is overly simplistic. Venom may be an important component of fire ant biology but it is clearly not the only explanation for their success or ecological impacts. More care is needed in the writing to highlight other causes and to acknowledge limitations.
439: There is no evidence that venom is more or less important than other aspects of fire ant biology.
441 – 443: This statement may be true for laboratory confrontations in glass vials, but in the field these two species are fairly closely matched in competitive abilities. This close matching is the conclusion of the paper (LeBrun et al 2007) that you are utilizing to support the reverse. This paper is mis-cited.
443 – 445: You previously cited data showing the reverse. S. geminata venom has greater insecticidal activity than S. invicta venom.
The Conclusions section provides a restatement of the text rather than providing major take aways ideas and areas for future research.
The sentence structure is often overly wordy, and areas of text could be tightened.
Reviewer 2 Report
This is a concise review on the venom alkaloids of fire ants, providing a useful compendium of what has been done in this topic so far. Overall, it is well prepared and worthy of publication.
Some aspects would merit , nonetheless, careful reconsideration from the authors.
(i) Firstly, it is unclear why the authors consider fire ants so special (as per statements in lines 7, 19, 37), like:
"Solenopsis fire ants (Hymenoptera: Formicidae) are special venomous ants."; "Fire ants in the genus Solenopsis (Hymenoptera: Formicidae) are especially venomous[...]"
Fire ants have been studied for about 100 years, due to the nuisance they can cause to humans. Yet, attention received by humans doesn't objectively mean they are "more venomous" than other ants, related or not, in a large perspective. Please be more specific. They are just more medically important, and widespread.
(ii) 'Alkaloid' is a loose term in chemistry, therefore a definition that applies to all cited compounds should be given;
(iii) Likewise, 'solenopsins' have been defined as term that seems not to include compounds as piperideines and pyridines; how can this be harmonized in this paper?
(iii) Mainly, the manuscript generalizes Solenopsis invicta as a standard for fire ants in general (which contain about 20 spp.). Much of what has been described for that species has not been tested for other studied species. For instance, gaster-flagging has not been thoroughly described in S. geminata or others as described in lines 86-88 for S. invicta. Likewise, lines 86-96 are about S. invicta only. Perhaps the scope of the manuscript should be declared to use S. invicta as a model, and make a better effort overall to emphasize where a particularity is known only of S. invicta.
(iv) There are some more obscure compounds reported for fire ants, such as piperidenes (example given below). Likely, at least brief comments about such compounds should be made.
https://www.pherobase.com/database/compound/compounds-detail-2me-6-pentadecenyl-6-piperidene.php
I hope the provided suggestions may prove useful to the authors.
Reviewer 3 Report
This is a nicely written review about Solenopsis invicta venom. Authors do a nice job of covering appropriate topics such as currently known alkaloids and general function. While helpful for those interested in these fire ants, I found the review lacking an important broader perspective. This is also reflected in the papers cited. I found many foundational and important papers missing. I mention a few other suggestions below.
Line 12: "nature" should be "natural"
There are a few cases where authors should replace "tail sting" with "stinger".
Authors should mention the diversity of ant genera (e.g., Monomorium, Myrmicaria, Erromyrma, Megalomyrmex, Chelaner) that have been found to have alkaloids in their venom and describe the shared (e.g., Pyrrolidine, piperidine) or differing (e.g., indolizine, indolizine) classes of alkaloids for broader context.
It should also be mentioned that frogs sequester alkaloids (solenopsin) thus the biological significance of venom alkaloids is much farther reaching than S. invicta. There are several very nice papers that review these topics and these should be cited with supporting papers with key findings. Please see Annual Review of Entomology, Journal of Chemical Ecology etc. for articles.
I appreciate the density of information about S. invicta but I can't emphasize enough that this work needs to be put in a broader ecological and evolutionary context so that others find it useful. A full paragraph added to the introduction would greatly enhance this review.
Line 40-41 is where similar alkaloids that are found in other ant species can be mentioned
Line 60: Wording "open or shut-off", seem to be missing a word here.
Line 97-99 Gaster flagging has also been observed in alkaloid dispensing Megalomyrmex. Cite and mention for broader context of this significant behavior.
Line 419 The ecological significance is extremely important and should be expanded. As it is, it gives the impression that little is known. Here you can also mention other work on other genera so that there is clear broad significance.
Line 470-471 This is really important point. Consider making a table or figure (i.e., visual synthesis) that emphasizes the summary of what you have reported here and above. More information should be provided for the following part as well (line 473-475).
Conclusion is lacking clearly framed hypotheses that can be tested by other researchers. This is another area where authors can advance the field by helping others build on what has been discovered in S. invicta.
English is fine
Round 2
Reviewer 3 Report
Authors have done a nice job addressing most of my concerns. While, I would have loved to hear what hypotheses the authors think are the most compelling to move this field forward (increasing the likelihood that this work would be cited), it seems they prefer to mostly leave this to the reader to think about. The table does help in this regard though. I think the text is fine as it is if the editors of Toxins agree.
Line 202: Insert "ant" after "fire"
Author Response
Authors have done a nice job addressing most of my concerns. While, I would have loved to hear what hypotheses the authors think are the most compelling to move this field forward (increasing the likelihood that this work would be cited), it seems they prefer to mostly leave this to the reader to think about. The table does help in this regard though. I think the text is fine as it is if the editors of Toxins agree.
We added one paragraph to move the topic forward in the "Conclusions" section.
Line 202: Insert "ant" after "fire"
Added.